# Performance Analysis of a Novel Hybrid Segmentation Method for Polycystic Ovarian Syndrome Monitoring

**DOI:** 10.3390/diagnostics13040750

**Published:** 2023-02-16

**Authors:** Asma’ Amirah Nazarudin, Noraishikin Zulkarnain, Siti Salasiah Mokri, Wan Mimi Diyana Wan Zaki, Aini Hussain, Mohd Faizal Ahmad, Ili Najaa Aimi Mohd Nordin

**Affiliations:** 1Department of Electrical, Electronic and Systems Engineering, Faculty of Engineering and Built Environment, Universiti Kebangsaan Malaysia, Bangi 43600, Selangor, Malaysia; 2Advanced Reproductive Centre, Department of Obstetrics and Gynaecology, Faculty of Medicine, Kuala Lumpur Campus, Universiti Kebangsaan Malaysia, Cheras 56000, Kuala Lumpur, Malaysia; 3Department of Electrical Engineering Technology, Faculty of Engineering Technology, Universiti Tun Hussein Onn Malaysia, Bandar Universiti Pagoh, KM1, Panchor, Pagoh 86400, Johor, Malaysia

**Keywords:** polycystic ovarian syndrome, image segmentation, follicle identification, Chan–Vese method, Otsu thresholding

## Abstract

Experts have used ultrasound imaging to manually determine follicle count and perform measurements, especially in cases of polycystic ovary syndrome (PCOS). However, due to the laborious and error-prone process of manual diagnosis, researchers have explored and developed medical image processing techniques to help with diagnosing and monitoring PCOS. This study proposes a combination of Otsu’s thresholding with the Chan–Vese method to segment and identify follicles in the ovary with reference to ultrasound images marked by a medical practitioner. Otsu’s thresholding highlights the pixel intensities of the image and creates a binary mask for use with the Chan–Vese method to define the boundary of the follicles. The acquired results were compared between the classical Chan–Vese method and the proposed method. The performances of the methods were evaluated in terms of accuracy, Dice score, Jaccard index and sensitivity. In overall segmentation evaluation, the proposed method showed superior results compared to the classical Chan–Vese method. Among the calculated evaluation metrics, the sensitivity of the proposed method was superior, with an average of 0.74 ± 0.12. Meanwhile, the average sensitivity for the classical Chan–Vese method was 0.54 ± 0.14, which is 20.03% lower than the sensitivity of the proposed method. Moreover, the proposed method showed significantly improved Dice score (*p* = 0.011), Jaccard index (*p* = 0.008) and sensitivity (*p* = 0.0001). This study showed that the combination of Otsu’s thresholding and the Chan–Vese method enhanced the segmentation of ultrasound images.

## 1. Introduction

Polycystic ovarian syndrome (PCOS) is an ovarian health condition of endocrine dysfunction that affects the health of women’s reproductive systems and is characterized by an ovary with a follicle count ≥12 and a size of 2 or 3 mm^3^ [1]. This syndrome produces side effects related to women’s overall health, such as abnormal hair growth, weight gain, irregular menstrual cycle, infertility and depression [2,3]. Almost 70% of women with PCOS suffer from infertility [4,5], and this condition is one of the common reasons for female infertility [6,7]. PCOS occurs in women during their reproductive years [8,9,10], between 18 and 45 years old.

The term ‘polycystic’ is attributed to the appearance of the ovaries, which are filled with multiple ovarian follicles of different sizes [11]. The etiology of this syndrome remains unknown and is currently being studied. Once a month, the ovaries produce follicles, known as oocytes, which will develop into an ovum during the ovulation phase. Ovulation is affected by hormones. The disruption of the hormones could cause imbalance, and ovulation will not happen. In every cycle, the ovary keeps producing follicles, and the follicles accumulate over time when hormonal imbalance disturbs the ovulation. Hence, PCOS occurs. Figure 1 shows a diagram to explain the differences between normal and polycystic ovaries. The normal ovary undergoes normal ovulation, and the polycystic ovary consists of multiple cysts due to follicle accumulation over time.

This syndrome is diagnosed via three approaches: medical imaging, pelvic exams, and blood tests. Medical imaging is one of the more prominent methods used by medical practitioners to diagnose health conditions [13,14]. Ultrasound imaging is commonly used to diagnose PCOS. Given that the most common PCOS characteristic is the presence of higher-than-normal follicle count, ultrasound imaging has been a significant help for medical practitioners to diagnose and supervise patients with PCOS. In addition to its usefulness, ultrasound imaging is inexpensive and portable [15]. 

In conventional PCOS diagnosis, medical practitioners freeze-frame the ultrasound image in a satisfactory position and measure the follicle size. The size and number of follicles can be measured manually or by marking the measurement to determine whether the follicles show PCOS characteristics. These steps are repeated for each follicle [16]. Figure 2a,b show two different ultrasound images of normal and polycystic ovaries [17].

The manual diagnosis of PCOS is laborious and prone to human error [18,19]. Furthermore, the drawback of ultrasound imaging is that low-quality images are produced [15]. Hence, medical practitioners face difficulty in calculating and measuring the follicles. Computer-assisted image analysis could improve clinical management to diagnose and monitor patients with PCOS [20]. Thus, researchers have explored various image processing methods to detect and measure follicles for PCOS diagnosis and monitoring.

The four main steps in image processing include pre-processing, segmentation, feature extraction and classification [15,21]. Image segmentation is a crucial and complicated step that determines the effectiveness of the analysis results [22]. This step helps provide information from the ultrasound image, such as the follicle count in the ovary and the size of the follicles. Image segmentation is the task of extracting identical or grouping pixels following either color, intensity, or texture [23]. The main aim of this step is to represent the image for easy information analysis. Therefore, researchers have been exploring and developing suitable segmentation methods for follicle identification in PCOS ultrasound.

In this study, the suggested method for image segmentation is by combining the Otsu’s thresholding method and the Chan–Vese active contour without using the edge method. The combination of the methods provides follicle segmentation with distinctive boundary detection using a quick and robust algorithm. While Otsu’s thresholding is able to provide a rough estimation of the segmentation based on finding the optimal threshold value for each class in the image, the Chan–Vese method refines the segmentation based on the initialization from the thresholding. Thus, the metric evaluation for this study was pixel-based evaluation. Furthermore, the performances of the methods were compared with follicles manually marked by a medical practitioner. Accuracy, Dice score, Jaccard index and sensitivity were used as evaluation metrics.

The paper is organized as follows. Section 2 presents a literature review of previous segmentation methods developed that are related to Otsu’s thresholding and the Chan–Vese method. The materials and methods are discussed in Section 3, where a flowchart of the proposed method is explained. Moreover, Section 3 explains the concept of Otsu’s thresholding and the Chan–Vese method with the evaluation metrics used for the study. Subsequently, in Section 4, the results of the two methods are presented and discussed with a comparison based on evaluation metrics, which are followed by a conclusion in Section 5.

## 2. Literature Review

Due to speckle noise disturbance in ultrasound images, researchers have been exploring the development of suitable methods to reduce or remove the speckle noise. The purpose of this is to help with the segmentation step while preserving information in the ultrasound. Different speckle-noise reduction methods have been implemented with different segmentation methods; however, no exclusive technique suitable for all segmentation methods has been employed. Furthermore, segmentation methods have been applied to help with speckle-noise reduction tasks, such as watershed segmentation, which was used as an image enhancement approach in the pre-processing step [24,25]. A few drawbacks have been found, such as the complexity of identifying small follicles and differentiating overlapping follicles with no distinctive boundaries [26,27]. These drawbacks could affect the diagnosis and treatment of patients.

Reviews and surveys of image segmentation methods have been conducted to determine a superior segmentation technique that is able to detect and calculate the number of follicles in the ovary [2,15,17,23,28,29,30,31,32,33,34,35]. One study [35] concluded that no exclusive method could be dominant considering the various characteristics of ultrasound images, especially ovarian ultrasound images. Thus, a competent image segmentation method depending on the image quality was proposed [36].

Thresholding is one of the simplest image segmentation methods [23,30]. Its basic principle is to classify the pixels according to intensities. The segmentation results are achieved by clustering or grouping the pixels with high intensities into one class [35]. There are two categories of thresholding method: local thresholding and global thresholding. The difference between these two thresholding approaches is the implementation of the threshold value. In the local thresholding approach, different thresholding values are applied to each pixel, depending on the local intensity distribution in its neighborhood. Meanwhile, global thresholding applies a single threshold value for the entire image. Otsu’s thresholding is one of the dominant thresholding methods [37] because it applies a threshold selection following the grey-level histograms of images. A comparison study between global thresholding and Otsu’s thresholding was conducted, and Otsu’s thresholding showed superiority in terms of sensitivity. Moreover, the detection size error of Otsu’s thresholding is higher than that of global thresholding [38]. Modified Otsu’s thresholding was recently implemented to binarize ultrasound images [37]. From the thresholding, the image then functions as a mask for active contour segmentation. In the study, the proposed method delivered satisfactory results for accuracy compared with other applied techniques.

Another renowned segmentation method is active contour, for medical imaging analysis. The proposal of active contour without edge helps identify regions of interest with or without distinctive walls [39]. The Chan–Vese method does not involve image smoothing even when the image is noisy. However, due to the inhomogeneity of ultrasound images, the Chan–Vese method alone performs poorly [40]. For this reason, Kumar and Srinivasan proposed an improved Chan–Vese with Split Bregman optimization to identify follicles with a diameter smaller than 2 mm. Another study was conducted by Lestari et al. using the morphology operation with the Chan–Vese model to compensate for shortcomings during segmentation and to obtain satisfactory results.

In 2019, ref. [37] proposed a combination of the active contour method with an improvised Otsu’s thresholding to automatically identify follicles. In their study, the active contour model required edge information prior to contouring the boundary of the follicles. Based on the results, the proposed method showed superiority in terms of accuracy identifying the number and size follicles existed in ovary. However, the study only showed superiority with respect to the accuracy of the proposed method. Further research is required to evaluate the segmentation method for medical images.

This study proposes a segmentation method that utilizes Otsu’s thresholding combined with the Chan–Vese method. The proposed method utilizes the concept in Otsu’s multilevel thresholding of finding the optimal threshold value to highlight the difference in pixel intensities in the images after the pre-processing step. The thresholded image provides an initial estimation of the follicles and wall boundaries in the ultrasound image. In the proposed method, the Chan–Vese method is then employed following Otsu’s thresholding to define the boundary of each follicle in the ultrasound. The combination of these two methods produces improved segmentation results, as the thresholding provides rough estimation of the follicles and boundaries and the active contour refines the follicles and boundaries in the output image.

## 3. Methodology

Ultrasound images of patients diagnosed with PCOS were used as input for the algorithm and were collected from Medically Assisted Conception (MAC), Hospital Canselor Tuanku Mukhriz (HCTM), Cheras, Kuala Lumpur, Malaysia. The ultrasound images were taken from patients with PCOS who were undergoing treatment and were monitored by medical practitioners. The follicle count in the ovary was manually calculated and determined by medical practitioners. These data, consisting of 20 ultrasound images, were the ground truth for this study.

This study aimed to identify the follicles in the ovary after the treatment was administered to the patients on the basis of the identified follicles manually marked by medical practitioners. The algorithm for both methods was written and run in MATLAB 2019a to execute the segmentation methods. The region of interest for each ultrasound image was extracted before inputting them into the algorithm. Images with identified follicles marked by medical practitioners were obtained and became the ground truth in the algorithm. 

Prior to the segmentation step, a pre-processing step was performed for each image to enhance the image contrast and remove speckle noise disturbance. The details of the pre-processing step in this study are explained in Section 3.1. Then, the segmentation step was carried out to identify and extract the follicles existing in the ultrasound images. The methods applied were the Classical Chan–Vese method and the proposed method combining Otsu’s thresholding with the Chan–Vese method. The method proposed in this study outputs the segmented follicles in ultrasound images as a binary image. Figure 3 shows the flow diagram of the method proposed in this study.

### 3.1. Pre-Processing Step

The ovary image is a small region of the total ultrasound image. The image was converted into a grayscale image to reduce computational procedures and simplify the algorithm. The pre-processing step was performed to improve the image contrast and enhance the region of interest for the segmentation step [41].

Different ultrasound images have been found to have different pixel intensities. In some ultrasound images, follicles that are close to the boundary may have light pixel intensities compared with the rest of the follicle [27]. This difference could lead to the light areas being neglected as part of the follicle. For some ultrasound images, the non-follicular area of the adjacent follicles may have the same intensity as the follicles. The similarity could include the non-follicular are and the adjacent follicles as one follicle [27]. 

Figure 4 and Figure 5 display different pixel intensities for different ultrasound images, with Figure 4 displaying an ultrasound image with a high intensity of dark pixels, while Figure 5 displays an ultrasound image with a high intensity of light pixels. The high intensity of dark pixels leads to poor image quality, and with speckle noise disturbance, some follicles boundaries have the same pixel intensity as the follicles. Meanwhile, the high intensity of light pixels emphasizes the boundary of follicles. Therefore, the aim of the pre-processing step is to enhance the visibility of the follicle boundaries, especially in the case of ultrasound images with a high intensity of dark pixels, for the next step. 

The ultrasound image in JPG format was encoded into MATLAB 2019a Workspace as an input image. The ultrasound image retrieved from HCTM was stored as RGB, although visually it is grayscale. Thus, it is necessary to convert the image into grayscale format for image processing, as grayscale images provide more information regarding the tissue structure. After the image was converted into grayscale, intensity transformation was conducted to increase the brightness and contrast of the image, especially around the region of interest, to overcome the pixel intensity problems. This step was performed because the saturated pixel intensity near the follicles’ boundary caused the invisibility of the edge boundary.

The median filter was implemented to reduce the speckle noise by moving through the image pixels and replacing each pixel with the median pixel value of its neighboring pixels. The median was calculated by arranging the pixel values from the window into the ascending order and then replacing the pixel considering the middle pixel value. Apart from reducing the speckle noise, the median filter can also smooth the edge of the region of interest and eliminate the noise from the image without reducing the accuracy [42]. A window size of 5 × 5 was applied. This window size is the optimal setting for reducing speckle noise [43] with performance in a short time [44]. The median filter was applied due to its gentle gradient on the follicles boundary and to preserve information [45]. Histogram equalization was then performed on the ultrasound image to compensate the contrast inequalities after median filter. The output from the pre-processing step provides a filtered image with reduced speckle noise and the presence of distinctive follicles in the ultrasound image. Figure 6 shows the steps of image pre-processing.

A medical expert was asked to mark the follicle boundary in the ultrasound image to be the ground truth, because this study will be implemented on local ultrasound images. Figure 7 shows the original ultrasound images that were collected by medical experts during patient monitoring, and Figure 8 shows the ultrasound images marked with red circles by medical experts to highlight the existing follicles.

### 3.2. Segmentation Step

After the pre-processing step, the filtered images were segmented with two segmentation methods, the classical Chan–Vese method and the proposed method, which is a combination of Otsu’s thresholding and the Chan–Vese method. The algorithm was written to simultaneously perform both methods. The proposed method employed Otsu’s multilevel thresholding with a threshold value of 4. In a previous study, the Otsu threshold was chosen manually due to the substandard pre-processing step [19]. In the present work, the threshold was fixed to a value of 4 in order to achieve desirable results in separating the follicles and wall boundaries on the basis of pixel intensities.

The binarized image from the thresholding method became the mask used in the Chan–Vese method to find the follicle boundary, which is an area with a high intensity of dark pixels. For this method, the iteration number was set as 500 for the fast and efficient segmentation processing of the image. The image outputs of both methods are in binary form. Figure 9 shows the segmentation steps.

#### 3.2.1. Otsu’s Thresholding

For the proposed method, Otsu’s thresholding, with a threshold level of 4, was introduced to the filtered image prior to the Chan–Vese method. The concept of Otsu’s thresholding is to find the optimal threshold value from the image histogram by minimizing its intra-class variance to segregate the image into two classes—foreground and background. For the calculation of variance, the pixels were divided into two classes: *b_0_*, with grey levels [0, *t* − 1]; and b1, with grey levels [*t*, …, *L* − 1].

The optimal threshold *t** was as written in Equation (1):(1)$$t^{*}=\arg\min\sigma_{\omega }^{2}\left(t\right).$$

The problem with using only one threshold in ultrasound images is that the pixel similarity in the grey areas, such as the boundary between two follicles or the wall of the ovary, could affect the thresholding result. Therefore, multilevel threshold with Otsu thresholding was implemented. The mathematical expression of global thresholding was as written in Equation (2):(2)G(x,y)=1 if F(x,y)≥t=0,
where *G* is the threshold image from *F*, the original image with threshold value *t*.

From Equation (2), the thresholding can be expressed as Equation (3):(3)$$b_{1}=q\;if\;0\;\leq q<t,\;b_{2}=q\;if\;t\;\leq q<L-1,$$
where *q* is one of the *x × y* pixels of the image, with *L* representing the grayscale *L* = {0, 1, 2, …, *L* − 1} and *b*_1_ and *b*_2_ referring to the classes where pixel *q* can be found. Corresponding to the rule for bilevel thresholding, the multilevel thresholding equation can be derived from the equation as shown in Equation (4):(4)$$b_{1}=q\;\mathrm{if}\;0\;\leq q<t_{1},\;b_{2}=q\;\mathrm{if}\;t\;\leq q<t_{2},\;b_{n}=q\;\mathrm{if}\;t_{n}\;\leq q<t_{n+1},\;b_{m}=q\;\mathrm{if}\;t_{m}\;\leq q<L-1,$$
where {*t*_1_, *t*_2_, *t*_3_,…, *t_n_*, *t*_*n*+1_, *t_k_*} define different threshold values.

In a state-of-the-art thresholding study, multilevel thresholding in Otsu’s thresholding was considered a desirable alternative for image segmentation for defined segmentation with less information loss from the image [46]. Afterward, the resultant image of Otsu’s thresholding became the mask for use in Chan–Vese segmentation.

#### 3.2.2. The Chan–Vese Method

The concept of the Chan–Vese method is to segregate the pixels using the active contour model based on curved evolution and deformation to identify the boundary of the region of interest without determining the edge boundary [39]. This method is formulated by minimizing the energy function for a domain image, Ω, of an image *I_0_* (*x,y*) by adding some regularization term, which is the length of parameterized curve of the image, *C*, and/or area of the region inside *C*. The formulation is displayed in Equation (5):(5)$$E\left(c_{1},c_{2},C\right)=\mu .\mathrm{length}\left(C\right)+\nu .\mathrm{Area}\left(\left(C\right)\right)+\lambda_{1}\int_{\mathrm{inside}\;\left(C\right)}{\left|I_{0}\left(x,y\right)-c_{1}\right|}^{2}dxdy+\lambda_{2}\int_{\mathrm{outside}\;\left(C\right)}{\left|I_{0}\left(x,y\right)-c_{1}\right|}^{2}dxdy,$$
where the fixed parameters are 𝜇 ≥ 0, 𝜈 ≥ 0 and 𝜆_1_,𝜆_2_ > 0. The energy function *c*_1_, *c*_2_ are the averages of the regions that approximate the image intensity inside and outside *C* expressed in the equation.

The energy is represented in the level set function as shown in Equation (6):(6)$$\{\begin{array}{c} C=\left\{I_{0}\left(x,y\right)\in \Omega \;:\;\phi \;\left(x\right)=0\right\}\\ \mathrm{inside}\left(C\right),c_{1}=\left\{I_{0}\left(x,y\right)\in \Omega \;:\;\phi \;\left(x\right)>0\right\}\\ \mathrm{outside}\left(C\right),c_{2}=\left\{I_{0}\left(x,y\right)\in \Omega \;:\;\phi \;\left(x\right)<0\right\} \end{array}.$$

The filtered images were segmented directly using the classical Chan–Vese method for the first data collection. For the second data collection, the binarized image from Otsu’s thresholding was segmented using the Chan–Vese method. In this study, segmentation was performed to determine the multiple existing follicles. Thus, the contour was grown inward. The number of iterations for the Chan–Vese method was set as 500 to ensure satisfactory results from both of the applied methods. The final image from the segmentation will be binary; the image marked by experts was binarized before comparison calculation. The implementation of the proposed method is summarized in the pseudo code algorithm below (Algorithm 1).
**Algorithm 1** Otsu’s thresholding and Chan–Vese segmentation method**Input image: original ultrasound image****Output image: segmented follicles image****Start**1. Read the image into the algorithm, identify the image properties2. Convert into grayscale image with pixels values of L [1, 2, 3, …, 255]3. Intensity transformation        Calculate the lower and upper bounds of the ultrasound image        Set target intensity of the lower and upper bounds.        Loop through all pixels and apply the intensty transformation.6. Median Filter        Calculate the median of the pixels based on 5 × 5 windows        Set the output image pixels to the median value8. Histogram equalization        Calculate the histogram of the image.        Calculate the cumulative distributive function of the histogram.        Map the pixels corresponding to the CDF11. Otsu’s Multilevel Thresholding        Calculate the threshold value for 4 levels.        Store the threshold value into the image13. Chan–Vese Active Contour        Initialize a curve level set function.        Compute mean intensity inside and outside the level set function        Compute the energy function and update the level set function        Repeat step 13 to 15 for 500 iterations14. Display the segmented image in binary**End**

### 3.3. Evaluation Metrics

Through segmentation, the performance of both methods was calculated and tabulated in reference to the ground truth images provided by medical experts. The tabulated evaluations were then discussed. For the assessment of the efficacy of the proposed method and for the comparison of outcomes between the two methods, the choice of the performance measurement was based on region information used to calculate the number of misclassified pixels [47]. Accuracy, Dice score, Jaccard index and sensitivity were used as evaluation metrics. Segmentation evaluation was compared between the classical Chan–Vese and the proposed method by calculating the *p*-value of the metrics between the two methods. If the *p*-value of the two methods is less than 0.05, then the proposed method shows a significant difference, thus exhibiting an improvement.

#### 3.3.1. Accuracy

Accuracy is achieved when the numerical values are calculated with the proportion of true positive results in the selected population [48,49]. In this study, accuracy was achieved when the proportion of follicles detected in the algorithm was comparable with the number of follicles counted by medical practitioners. Higher values of accuracy show that the segmentation method is able to identify the follicles effectively, while lower accuracy values indicate that the segmentation performed poorly.

Accuracy was formulated as shown in Equation (7):(7)$$\mathrm{accuracy}=\frac{\mathrm{TN}\;+\;\mathrm{TP}}{\mathrm{FN}\;+\;\mathrm{FP}+\;\mathrm{TN}+\;\mathrm{TP}},$$
where TP, TN, FP and FN are true positive, true negative, false positive and false negative, respectively. True positive refers to follicles that were segmented correctly by the segmentation method on the basis of the follicles segmented by medical practitioner, while true negative corresponds to non-follicular area that was detected as non-follicular by both the segmentation method and the medical practitioner. False positive and false negative are the segmented follicles and non-follicular area determined by the segmentation method that were not segmented and detected by the medical practitioner. These parameters are the basic statistical indices used for evaluating segmentation results based on ground truth [50].

#### 3.3.2. Dice Score and Jaccard Index

The Dice score and Jaccard index calculate the overlapping area of sample sets from ground truth and segmented images. The Dice score was used to quantify the similarity of two samples. With *J* as the segmented region that requires evaluation, and *I* as the ground truth for the image, the Dice score can be expressed as shown in Equation (8):(8)$$\mathrm{dice}=\frac{2\;\left|J\;\cap^{\;}I\right|}{\left|J\right|+\left|I\right|},$$
where |.| are the cardinalities of the sets, i.e., the number of elements of each set in images *I* and *J*. The intersect operator in Equation (8) indicates the pixels sets in both *I* and *J*. The Dice score has a value between 0 and 1; the higher the value is, the better the segmentation result [51,52].

The Jaccard index corresponded to the Dice score in Equation (8):(9)$$\mathrm{Jaccard}=\frac{\mathrm{dice}}{2-\mathrm{dice}}$$

Thus, the equation of the Jaccard index was written as shown in Equation (9):(10)$$\mathrm{Jaccard}=\frac{\;\left|J\;I\right|}{\left|J\right|+\left|I\right|-\left|J\;I\right|}$$

The Jaccard index is the calculated area of overlap between the segmented and the ground truth areas divided by the unity of the area between the segmented area and ground truth. The evaluation of the Jaccard index metric falls in the range between 0 and 1 (0, no overlap; 1, perfectly overlapping segmentation). The higher the value, the better the segmentation results [51,52].

#### 3.3.3. Sensitivity

Sensitivity measures the number of actual positives and negatives correctly identified during segmentation [49]. It measures the proportion of segmented follicles that are correctly segmented according to the ground truth. The equation of sensitivity can be written as shown in Equation (10).
sensitivity = TP/(TP + FN)(11)

Higher sensitivity values show that the segmentation method was able to segment follicles in the ultrasound image correctly, according to the ground truth. Meanwhile, lower sensitivity values mean that the segmentation method poorly segmented the follicles present.

## 4. Results and Discussion

Figure 10 presents the images in the pre-processing step: intensity transformation, median filter, and histogram equalization. Figure 10a shows the ultrasound image before any pre-processing steps were taken. Speckle noise disturbances were found throughout the image. As a result, the image was of poor quality, with some parts of the wall lining seeming to portray the existence of follicles.

Contrast enhancement adjusts the brightness and darkness of the ultrasound image to improve the image [53] and help reduce the speckle noise. In Figure 10b, the difference between the image pixels in the light and dark areas was improved by contrast enhancement. Furthermore, less speckle noise was observed the image. However, the leftover speckle noise residue, especially in the ovary wall lining, could still affect the segmentation process. Hence, Figure 10c shows the image after median filtering. The median filter smoothened the image and reduced the speckle noise disturbances while keeping the edge of the follicles’ boundaries. Figure 10d shows the image after histogram equalization, showing that equalization helped reduce speckle noise disturbances and visualize great separation between the dark and light areas of the image.

Figure 11 presents a comparison of the Otsu’s thresholding of the original image before and after the pre-processing steps. In Figure 11a, the boundaries of the follicles are poorly differentiated compared with those in Figure 9b. With the help of the pre-processing step, the pixel intensity was normalized to achieve good foreground and background separation. The pixel intensity in the rear of the ultrasound was similar to that in the follicle area, leading to a misidentification of follicles.

Figure 11b shows that Otsu’s thresholding managed to separate the follicles’ boundaries from the ovary without speckle noise disturbance. Thresholding created homogeneity in the ultrasound image for the Chan–Vese method due to its capability to effectively segment homogenous images.

Figure 12 presents a comparison of the classical Chan–Vese and proposed methods with the follicles marked by medical experts. The original ultrasound image is shown in Figure 12a. The image is of poor quality due to speckle noise disturbances, and the boundaries between follicles are poorly visible. In Figure 12b, the red circles are follicles marked by medical experts, showing which of the dark regions are the follicles. Figure 12c shows that the classical Chan–Vese method was able to recognize all small-sized follicles in the ultrasound image. The follicles close to the rear of the ultrasound had a high dark intensity, similar to their boundaries. Otherwise, the follicles detected were relatively small in size due to the inhomogeneity of the ultrasound image. By contrast, the Chan–Vese method is suitable for homogenous images [40]. Figure 12d shows that the follicles were recognized according to the area marked by medical experts. Even though the detected follicles were nearly the same size as the marked ones, some other areas were also detected as follicles. These areas were capillaries in the ovary ultrasound that had similar intensities to the follicles in the ovary.

Figure 13 shows another ultrasound image with three different segmentations: manual segmentation, classical Chan–Vese, and the proposed segmentation method. This ultrasound image has better image quality and lesser speckle noise reduction than the others. Figure 13a shows the original ultrasound image. Compared with Figure 12a, the speckle noise disturbances in Figure 13a do not affect the ultrasound. Moreover, Figure 13b shows the binarized follicles marked by a medical practitioner to provide a ground truth image for this study. Figure 13c,d show the difference in segmentation results. The follicles in Figure 13c, segmented using the classical Chan–Vese method, are poorly identified. The follicle boundaries may have been poorly identified during pre-processing, and without using the Otsu thresholding method, the boundary pixels have affected the segmentation results. This poor boundary detection will affect size measurements. The segmented image in Figure 13d is the closest to the manually segmented image. With the help of Otsu thresholding, the pixels have been segregated clearly to show the follicle area.

A comparison of the ground truth and segmented images is shown in Figure 12 and Figure 13. The images obtained using classical Chan–Vese method and the proposed method in Figure 12c,d and Figure 13c,d show that some false follicles were detected, because the pixel intensities in the particular area mimicked the intensities of PCOS follicles. The quantitative values of image segmentation were calculated in order to compare the effectiveness of the methods.

Table 1 presents the value for different evaluation metrics, namely, accuracy, Dice score, Jaccard index and sensitivity, for the segmentation performance of the classical Chan–Vese and proposed methods. The accuracy values for the segmentation methods were similar. Overall, the accuracy of the proposed method had an average of 0.89 ± 0.04, which was 3.25% slightly higher than that of the classical Chan–Vese method, with a score of 0.86 ± 0.06. For different evaluation metrics, the proposed method showed superior performance in terms of Dice Score, Jaccard index and sensitivity. The average Dice score value for the classical Chan–Vese was 0.67 ± 0.12, which was 10.58% lower than that of the proposed method, at 0.77 ± 0.11. The bold of the value is to highlight higher result value.

Given that the Jaccard index corresponds to the Dice score value, the proposed method also showed superior performance, with an average of 0.64 ± 0.14, which was 12.88% higher than that achieved using the classical Chan–Vese method, at 0.51 ± 0.14. For sensitivity, all images showed high values with the proposed method. The average sensitivity of the proposed method was 0.74 ± 0.12, which was 20.03% higher than that of the classical Chan–Vese method, at 0.54 ± 0.14.

Figure 14 displays a bar chart comparing the average values of the evaluation metrics between the classical Chan–Vese and the proposed methods. According to the bar chart, the accuracies of both methods were similar, with the proposed method showing an accuracy slightly superior to that of the classical Chan–Vese method, with an average difference of 0.0325. For the Dice score, Jaccard index and sensitivity metrics, the proposed method possessed a superior sensitivity. The differences between the average Dice score, Jaccard index and sensitivity were 0.1058, 0.1289 and 0.2003, respectively.

According to the bar chart, the accuracy of both methods was high, almost reaching 1, which is regarded as a good value for segmentation. However, the accuracy results deal with true segmentation, a reference for which is non-existent in medical cases. Even if the ground truth images were marked by medical practitioners, because the images were manually segmented, some of the truth might be lost due to human error. The accuracy value should ideally calculate the difference between real true segmentation and the experimental segmentation [54,55], and is thus not highly reliable. For segmentation evaluation, other evaluation metrics must be considered. Owing to the different dark and light pixel intensities of different ultrasound images, some images showed better segmentation when using the classical Chan–Vese method than with the other. However, this was the case in 3 out of 20 ultrasound images.

On the basis of the tabulated data, box-and-whisker plots were drawn to compare the classical Chan–Vese method and Otsu’s thresholding with the Chan–Vese method. Figure 15 shows the box-and-whisker plot with significant differences for each metric of evaluation. In each of the box-and-whisker plots, the *p*-value for each metric was calculated and shown. Our results showed that the average of three of the metrics significantly improved (*p* < 0.05), namely, Dice score (*p* = 0.011), Jaccard index (*p* = 0.008) and sensitivity (*p* = 0.0001). However, a comparison of accuracy between the classical Chan–Vese and proposed methods did not show any significance (*p* > 0.05).

The most commonly used metrics for medical image segmentation is Dice score and Jaccard index, because they are straightforward in calculating the overlapping area between the ground truth and the predicted area after segmentation. Given that this study proposed a new method for image segmentation for PCOS ultrasound images, the images marked by medical practitioners served as the ground truth to aid with the calculation of the evaluation metrics. The significant increase in the Dice score and Jaccard index of the proposed method indicate its superiority for image segmentation compared with the classical Chan–Vese method.

## 5. Conclusions

In this study, the different properties of ultrasound images were highlighted. Different ultrasound images have different pixel intensities and qualities, and these attributes are affected by the probes used and the condition of the follicles. Researchers have explored and developed different approaches to segmenting follicles for diagnosis and monitoring purposes.

In this study, the proposed method combined Otsu’s thresholding and the Chan–Vese method to achieve improved image segmentation results. Two methods of segmentation, the classical Chan–Vese segmentation and the proposed method, were applied and compared in order to evaluate their performance. Their segmentation performance was evaluated on the basis of metrics of accuracy, Dice score, Jaccard index and sensitivity. According to the evaluation metrics, the proposed method was superior in terms of sensitivity, Dice score and Jaccard index. Even though the respective accuracies of both methods were similar, the other metrics demonstrated a great difference between the two methods with respect to their effectiveness for image segmentation. The classical Chan–Vese method alone was found to be unsuitable for segmentation due to the inhomogeneity of ultrasound images. Combining different segmentation methods, such as Otsu’s thresholding and the Chan–Vese method, can be used to improve the state-of-the-art segmentation methods of ultrasound images, especially when performing follicle identification.

This study focused on the segmentation step, without feature extraction and classification. Therefore, the results of this study are limited to the segmentation results achieved using the methods applied. Moreover, the data collected were images of patients who had been diagnosed with PCOS and were undergoing treatment. Diagnosis processes were not needed. Further research is needed to improve the method’s effectiveness and suitability for different ultrasound image properties to be applied for the next image processing steps, namely, feature extraction and classification. Furthermore, the sensitivity metrics of the method could be improved to achieve the standard sensitivity acceptable by the medical community for future implementation. Different evaluation metrics, such as Dice Score and Jaccard index, should be explored in more detail in order to optimize the method proposed in this study.

A suggestion for future work is to explore and develop the proposed method by acquiring ultrasound images from patients that have not yet been diagnosed with PCOS. This study could explore feature extraction from post-segmented images to diagnose PCOS. Another future work is to extend the research to classification by measuring the size, shape and volume of the follicles segmented. This extended study would require medical information such as menstrual history and hormonal levels.

## Figures and Tables

**Figure 1 diagnostics-13-00750-f001:**
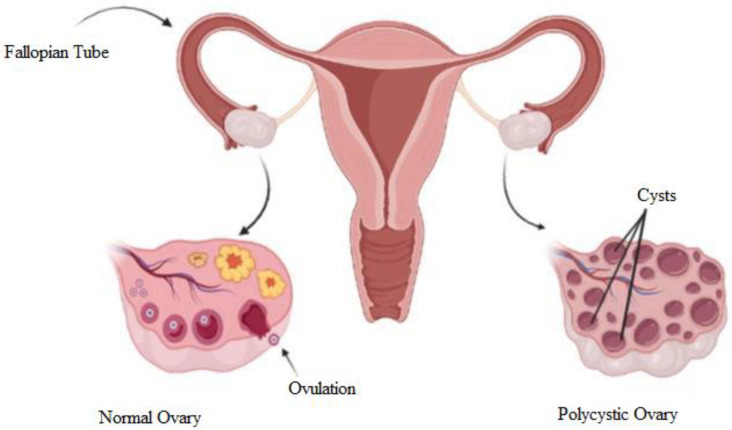
Diagram of normal and polycystic ovaries [12].

**Figure 2 diagnostics-13-00750-f002:**
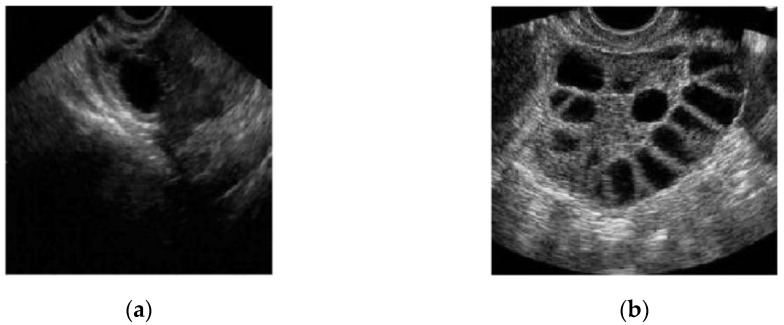
Ultrasound image of (**a**) a normal and (**b**) a polycystic ovary [17].

**Figure 3 diagnostics-13-00750-f003:**
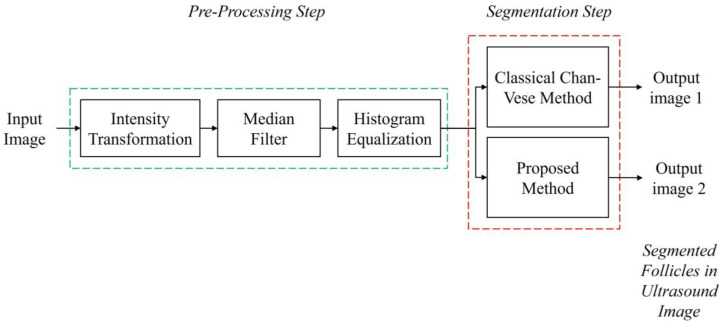
Flow diagram of the study.

**Figure 4 diagnostics-13-00750-f004:**
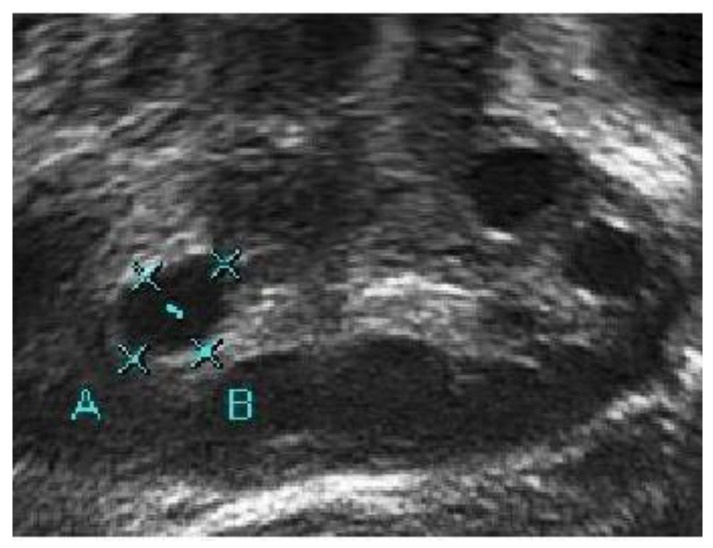
Ultrasound image with high intensity of dark pixels.

**Figure 5 diagnostics-13-00750-f005:**
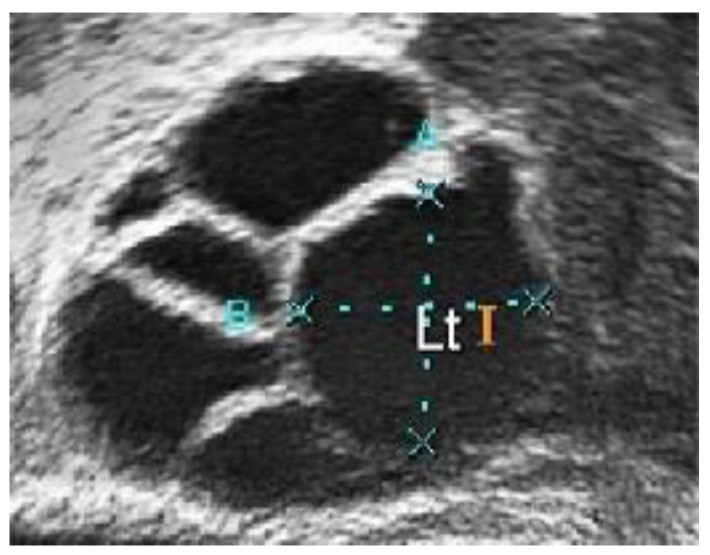
Ultrasound image with high intensity of light pixels.

**Figure 6 diagnostics-13-00750-f006:**
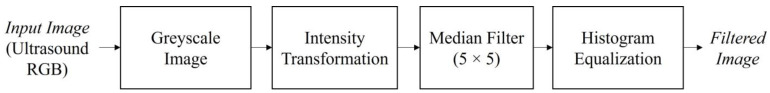
Pre-processing steps of the ultrasound image before the image segmentation step.

**Figure 7 diagnostics-13-00750-f007:**
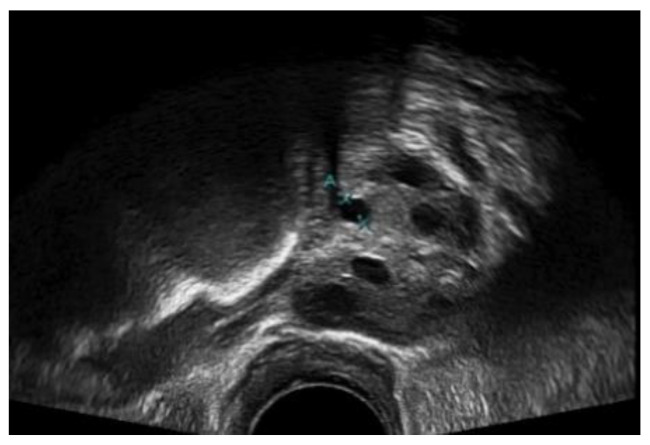
Original ultrasound acquired from the HCTM MAC unit.

**Figure 8 diagnostics-13-00750-f008:**
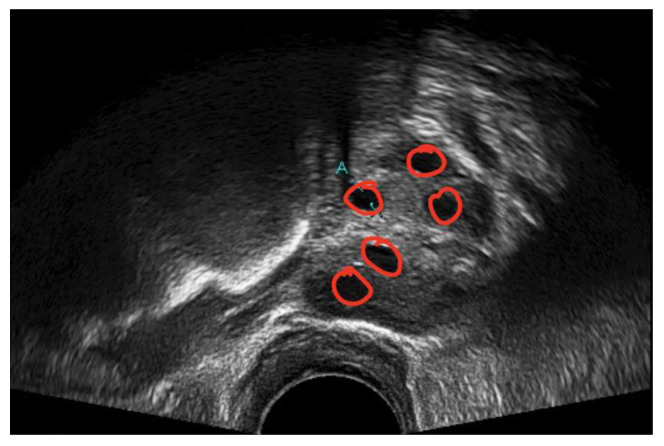
Image of identified follicles marked by medical expert.

**Figure 9 diagnostics-13-00750-f009:**
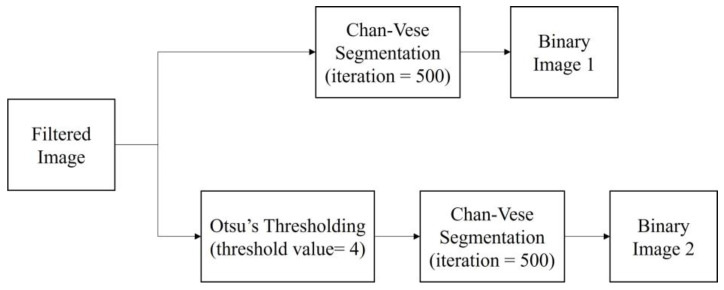
Segmentation methods applied in the study.

**Figure 10 diagnostics-13-00750-f010:**
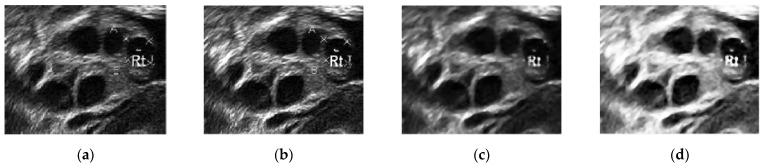
(**a**) Original ultrasound image, (**b**) ultrasound image after contrast enhancement, (**c**) ultrasound image with median filter, and (**d**) ultrasound image after histogram equalization.

**Figure 11 diagnostics-13-00750-f011:**
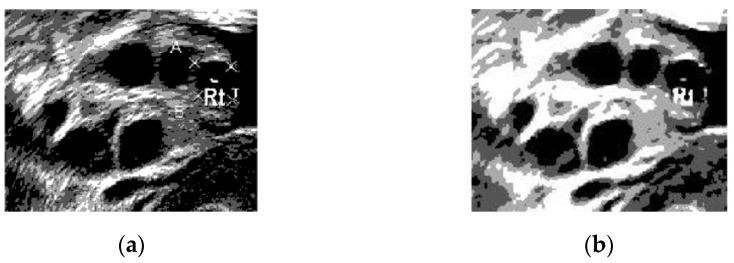
(**a**) Original ultrasound image, (**b**) ultrasound image after Otsu’s thresholding with a grayscale threshold level of 4.

**Figure 12 diagnostics-13-00750-f012:**
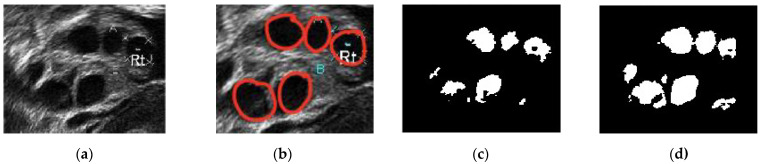
Comparison images of different segmentation methods. (**a**) Original ultrasound image; (**b**) follicles marked by medical expert; (**c**) classical Chan–Vese method; and (**d**) Otsu thresholding and the Chan–Vese method.

**Figure 13 diagnostics-13-00750-f013:**
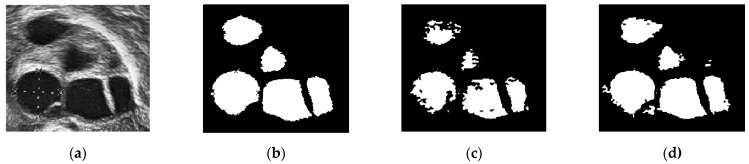
Comparison of different segmentation methods. (**a**) Original ultrasound image; (**b**) manual segmentation; (**c**) classical Chan–Vese method; and (**d**) Otsu thresholding and the Chan–Vese method.

**Figure 14 diagnostics-13-00750-f014:**
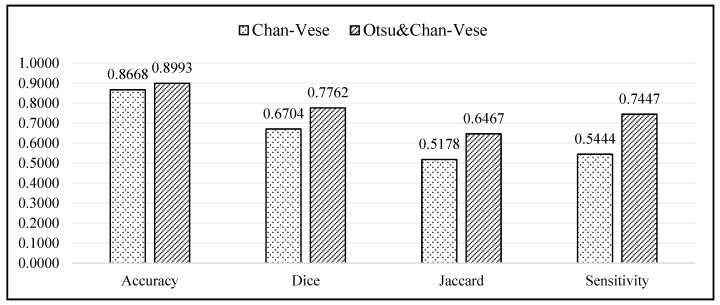
Bar chart comparing the accuracy, Dice Score, Jaccard Index, and sensitivity of the classical Chan–Vese method and the proposed method.

**Figure 15 diagnostics-13-00750-f015:**
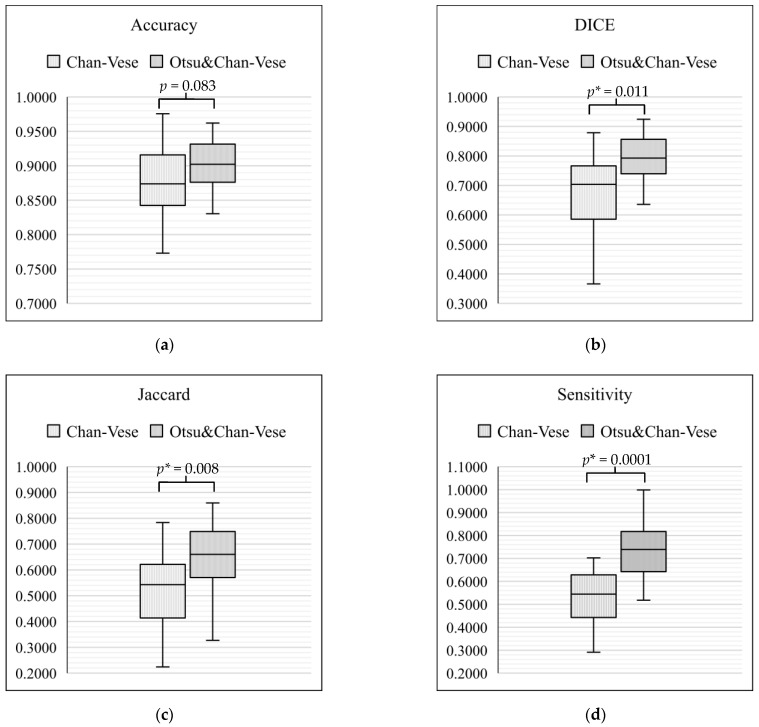
Box-and-whisker plot comparison between the classical Chan–Vese and Otsu’s thresholding with the Chan–Vese method. (**a**) Box-and-whisker plot for accuracy; (**b**) box-and-whisker plot for Dice Score; (**c**) box-and-whisker plot for Jaccard value; (**d**) box-and-whisker plot for sensitivity. * means *p*-value with the less than 0.05.

**Table 1 diagnostics-13-00750-t001:** Comparison of the accuracy, Dice Score and Jaccard index for the classical Chan–Vese and proposed methods.

Image	Accuracy		Dice Score		Jaccard Index		Sensitivity	
	Chan–Vese	Otsu and Chan–Vese	Chan–Vese	Otsu and Chan–Vese	Chan–Vese	Otsu and Chan–Vese	Chan–Vese	Otsu and Chan–Vese
Image 1	0.7900	**0.8303**	0.5869	**0.7275**	0.4154	**0.5717**	0.4268	**0.6481**
Image 2	0.8891	**0.9231**	0.7583	**0.8568**	0.6107	**0.7495**	0.6116	**0.8099**
Image 3	0.7064	**0.8313**	0.5152	**0.7690**	0.3470	**0.6247**	0.3472	**0.6249**
Image 4	0.8442	**0.9007**	0.7051	**0.8132**	0.5445	**0.7111**	0.5445	**0.7149**
Image 5	0.7951	**0.8500**	0.6082	**0.7498**	0.4370	**0.5598**	0.4421	**0.6247**
Image 6	0.8446	**0.9068**	0.7242	**0.8561**	0.5677	**0.7484**	0.5727	**0.7775**
Image 7	0.8778	**0.9112**	0.7570	**0.8385**	0.6090	**0.7219**	0.6150	**0.7445**
Image 8	**0.9756**	0.8935	**0.8788**	0.6357	**0.7838**	0.4660	0.9492	**0.9986**
Image 9	0.8482	**0.9477**	0.6366	**0.9013**	0.4669	**0.8203**	0.4685	**0.8406**
Image 10	0.9139	**0.9607**	0.8121	**0.9244**	0.6837	**0.8595**	0.6855	**0.8865**
Image 11	0.8932	**0.9621**	0.7023	**0.9190**	0.5412	**0.8502**	0.5448	**0.9299**
Image 12	0.8444	**0.8563**	0.5811	**0.7237**	0.4095	**0.5670**	0.4426	**0.7713**
Image 13	**0.9215**	0.8827	**0.7904**	0.7721	**0.6535**	0.6288	0.6858	**0.9208**
Image 14	0.8913	**0.9261**	0.7077	**0.8175**	0.5476	**0.6913**	0.5557	**0.6992**
Image 15	0.8697	**0.9030**	0.6321	**0.7503**	0.4621	**0.6004**	0.4626	**0.6024**
Image 16	0.8370	**0.8993**	0.4995	**0.7437**	0.3329	**0.5920**	0.3339	**0.6001**
Image 17	0.7730	**0.7915**	0.3663	**0.5284**	0.2242	**0.3590**	0.2910	**0.5181**
Image 18	0.9403	**0.9488**	0.7989	**0.8357**	0.6652	**0.7178**	0.6683	**0.7332**
Image 19	0.9488	**0.9598**	0.8240	**0.8680**	0.7007	**0.7667**	0.7026	**0.7752**
Image 20	**0.9322**	0.9014	**0.5231**	0.4930	**0.3542**	0.3271	0.5373	**0.6740**
Average	0.8668	**0.8993**	0.6704	**0.7762**	0.5178	**0.6467**	0.5444	**0.7447**

## Data Availability

The data collected are under ethical approval which is only applicable for this study and shall not be publicly available.

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
