# Peer review of "Performance Analysis of a Novel Hybrid Segmentation Method for Polycystic Ovarian Syndrome Monitoring"

_diagnostics, 2023, doi:10.3390/diagnostics13040750_

Round 1
Reviewer 1 Report
The following points caused the major revision of the article:
- Chan-Vese and Otsu’s methods are quite old and prominent methods. Where is your contribution?
- The title needs revision. The word combinatorial is not suitable in this situation. I can suggest titles like: “Performance Analysis of a Novice Hybrid Image Segmentation Method for Polycystic Ovarian Syndrome Monitoring and Diagnosis”
- The article needs to be thoroughly addressed for typos and grammatical errors.
- The organization of the article is wrong. Correct it.
- Why the standard deviation of your results is too high? 0.74 ± 0.12
- How 20.03% is the gap? The sensitivity is not a prominent measure in comparison to the dice coefficient and Jaccard index.
- 20 ultrasound images are insufficient for publication. The data should be publicly available. The results must be reproducible by others.
- Try to avoid writing immature wordings like: “The algorithm of both methods were written and run 141 in MATLAL 2019a to execute the segmentation methods.:”. Instead, give a pseudo-code of your work that we shall check and comment on.
- Figure 3 is faulty and needs to be corrected.
- Lines 154 to 160 need correction.
- Figures 4 and 5 are not properly explained. Write details about them.
- Most ultrasound images are black and white. You mentioned RGB ultrasound images in Fig. 6. Although mostly they are gray-level images, colors are used to refer to some other variations. Explain it.
- The last step in Fig 6 is missing in the text
- Segmentation is not clear 195-200. Throughout the article, you have been changing the system of segmentation. Why not decide on a single way out?
- Figure 9 is wrong. Correct it according to your text.
- Evaluation metrics are written in an extremely poor manner.
- The segmentation results are poor having high variance. Another problem is that the number of trial cases is limited and not open to the public.
Reviewer 2 Report
This paper studies the Performance Evaluation on Combinatorial Image Segmentation Method for Polycystic Ovarian Syndrome Monitoring. Some weaknesses should be addressed in this paper. Therefore, I suggest the authors resubmit it after a major revision. My suggestions are as follows:
1. Discuss the study's limitations and future research suggestions.
2. I strongly suggest that the paper be proofread and reread meticulously again, particularly regarding the spelling and grammatical mistakes.
3. In Fig3, you provided a flowchart to explain the Evaluation on Combinatorial Image Segmentation Method. This section must provide a concise and clear explanation of the suggested approach. Although the flowchart is beneficial, it’s also important to outline the methodology behind this new approach.
4. Divide the introduction into two parts of introduction and literature review. Add a literature review section after the introduction.
5. Please outline the structure of your paper at the end of the introduction with more details and explanations of subsections.
6. I suggest that you update section 2.1 so that the illustration used in the Pre-Processing Step subsection should be more readable.
7. Please clarify the definitions for all equations. Meanwhile, all equations are the same number of 1 and 2. It should be in order of 1,2,3,4,5…..
8. It is necessary to include additional information for the Evaluation Metrics in line 249.
9. Following the mathematical model is difficult due to a few notational mistakes.
10. To improve your related works, consider the following related references.
- Sreejith S, Nehemiah HK, Kannan A. A clinical decision support system for polycystic ovarian syndrome using red deer algorithm and random forest classifier. Healthcare Analytics. 2022 Nov 1; 2:100102.
- Tiwari S, Kane L, Koundal D, Jain A, Alhudhaif A, Polat K, Zaguia A, Alenezi F, Althubiti SA. SPOSDS: A smart Polycystic Ovary Syndrome diagnostic system using machine learning. Expert Systems with Applications. 2022 Oct 1; 203:117592.
- A novel machine learning approach combined with optimization models for eco-efficiency evaluation. Applied Sciences. 2020 Jul 28;10(15):5210.
- Zhang X, Liang B, Zhang J, Hao X, Xu X, Chang HM, Leung PC, Tan J. Raman spectroscopy of follicular fluid and plasma with machine-learning algorithms for polycystic ovary syndrome screening. Molecular and cellular endocrinology. 2021 Mar 1; 523:111139.
- Developing a novel integrated generalised data envelopment analysis (DEA) to evaluate hospitals providing stroke care services. Bioengineering. 2021 Dec 10;8(12):207.
- Kazemi M, McBreairty LE, Chizen DR, Pierson RA, Chilibeck PD, Zello GA. A comparison of a pulse-based diet and the therapeutic lifestyle changes diet in combination with exercise and health counselling on the cardio-metabolic risk profile in women with polycystic ovary syndrome: a randomized controlled trial. Nutrients. 2018 Sep 30;10(10):1387.
- Azouz AA, Ali SE, Abd-Elsalam RM, Emam SR, Galal MK, Elmosalamy SH, Alsherbiny MA, Hassan BB, Li CG, El Badawy SA. Modulation of steroidogenesis by Actaea racemosa and vitamin C combination, in letrozole induced polycystic ovarian syndrome rat model: promising activity without the risk of hepatic adverse effect. Chinese Medicine. 2021 Apr 29;16(1):36.
- A novel hybrid parametric and non-parametric optimisation model for average technical efficiency assessment in public hospitals during and post-COVID-19 pandemic. Bioengineering. 2021 Dec 27;9(1):7.
- Zhang J, Sun Z, Jiang S, Bai X, Ma C, Peng Q, Chen K, Chang H, Fang T, Zhang H. Probiotic Bifidobacterium lactis V9 regulates the secretion of sex hormones in polycystic ovary syndrome patients through the gut-brain axis. Msystems. 2019 Apr 30;4(2):e00017-19.
- Schubert M, Mettler L, Deenadayal Tolani A, Alkatout I. Fertility Preservation in Endometrial Cancer—Treatment and Molecular Aspects. Medicina. 2023 Jan 24;59(2):221.
- Scannell N, Moran L, Mantzioris E, Cowan S, Villani A. Efficacy, feasibility and acceptability of a Mediterranean diet intervention on hormonal, metabolic and anthropometric measures in overweight and obese women with polycystic ovary syndrome: study protocol. Metabolites. 2022 Mar 31;12(4):311.
Round 2
Reviewer 1 Report
The paper may be accepted in current form.
Reviewer 2 Report
This version is available for the publication.